# Anorexia Nervosa—What Has Changed in the State of Knowledge about Nutritional Rehabilitation for Patients over the Past 10 Years? A Review of Literature

**DOI:** 10.3390/nu13113819

**Published:** 2021-10-27

**Authors:** Katarzyna Jowik, Marta Tyszkiewicz-Nwafor, Agnieszka Słopień

**Affiliations:** Department of Child and Adolescent Psychiatry, Poznan University of Medical Sciences, 60-572 Poznan, Poland; malamt@gmail.com (M.T.-N.); agaslopien@ump.edu.pl (A.S.)

**Keywords:** anorexia nervosa, refeeding, medical assessment, treatment, nutritional rehabilitation, supplementation, microbiome

## Abstract

Anorexia nervosa (AN) is a psycho-metabolic disorder with a high risk of somatic complications such as refeeding syndrome (RFS) and carries the highest mortality rate of all psychiatric illnesses. To date, the consensus on the care for patients with AN has been based on recommendations for a combination of alimentation and psychotherapy. It is important to establish an initial caloric intake that will provide weight gain and minimize the risk of complications in the treatment of undernourished patients. Research over the past few years suggests that current treatment recommendations may be too stringent and should be updated. The aim of this paper is to systematize the current reports on nutritional rehabilitation in AN, to present the results of studies on the safe supplementation of patients and its potential impact on improving prognosis and the healing process. This review of literature, from 2011–2021, describes the changing trend in the nutritional protocols used and the research on their efficacy, safety, and long-term effects. In addition, it presents previous reports on the potential benefits of introducing vitamin, pro-and prebiotic and fatty acid supplementation.

## 1. Introduction

Anorexia nervosa (AN) is a complex and chronic disease of unknown etiology, often causing relapses and leading to disability. It is characterized by abnormal eating behaviors, an excessive drive to lose weight, and distorted body image. Two subtypes of eating behavior in patients with AN are typically described. In restricting AN, patients lose weight solely through diet and exercise, whereas patients with binge-eating/purging AN restrict food intake and exercise, as well as periodically engage in paroxysmal overeating and/or purging.

AN is often associated with disease denial and resistance to treatment. Continued restrictive eating and undernutrition can cause dysfunction of nearly every system, including cardiovascular, endocrine, gastrointestinal, or gynecological complications and other metabolic changes, threatening potentially serious medical complications [1]. Most complications resolve with normalization of body weight and/or cessation of purging behavior. Therefore, restoration of nutrition and subsequent weight restoration is a key component of treatment to avoid serious physical complications and to improve psychopathological symptoms and cognitive function.

Many publications are available yielding sometimes divergent hypotheses on the optimal nutritional rehabilitation of patients with AN hospitalized in psychiatric wards. This has hindered the development of evidence-based guidelines for nutritional rehabilitation in AN. This review is intended to summarize the literature from the past 10 years regarding the most optimal nutritional strategies necessary for weight restoration and the nutrients to consider in supplementation for individuals in AN nutritional rehabilitation. It also highlights the urgent need to update and standardize international therapeutic standards relating to AN.

## 2. Calories and Nutrients in the Re-Nutrition Plan

Treatment guidelines published by medical societies in different countries vary considerably—the German, Danish and World Federation of Societies for Biological Psychiatry (WFSBP) standards do not contain specific recommendations for energy intake during AN therapy [2,3,4]. The MARSIPAN guidelines recommended starting with a caloric intake of 20 kcal/kg/day, and even 5–10 kcal/kg for patients at high risk of metabolic complications, such as refeeding syndrome (RFS) [5]. A different approach to nutritional rehabilitation is taken by the American Psychiatric Association (APA) recommending starting refeeding at 30 to 40 kcal/kg/day (average 1000–1600 kcal/day), with a feeding plan that can be modified to 70–100 kcal/kg/day [6]. It also points out how important it is to encourage patients with AN to expand their dietary choices to minimize their severely limited range of acceptable foods.

There is an ongoing debate about how to proceed with the re-nutrition of severely undernourished patients. Publications and recommendations written before 2015 were dominated by the direction of cautious introduction and caloric increase in the initial stage of nutritional rehabilitation. The first publications testing the hypothesis of higher initial caloric intake confirmed that such a strategy accelerates weight restoration and total hospitalization time for patients with AN [7,8]. Newer research has increasingly challenged the “start low, go slow” approach and promoted higher calories from the onset of refeeding [9,10,11,12]. Especially since even with low caloric intake, RFS is still documented. The Royal College of Psychiatrists standards (NICE 2016) assume that setting the required caloric intake during nutritional rehabilitation must take into account restoration of nutritional status as soon as possible while maintaining the physiological limitations associated with undernutrition and the patient’s mental resources [13].

In a recent randomized trial by Golden and Garber involving 111 patients with AN, participants were randomly assigned to either the HCR group (higher calorie refeeding, taking 2000 kcal per day, increasing 200 kcal per day) or the LCR group (lower calorie refeeding, taking 1400 kcal per day, increasing 200 kcal every other day). Rates of medical readmissions, number and duration of readmissions, and changes in global EDE-Q eating disorder questionnaire scores over time did not differ between groups. The finding that clinical remission and medical readmission did not differ at one year, combined with the final treatment outcomes, support the greater efficacy of the higher initial calorie approach compared to a more cautious approach. The results presented here allay previous concerns of a large body of clinicians that faster weight gain in patients hospitalized for AN could be associated with a higher risk of re-hospitalization [14,15].

Similarly, in the Davis experiment, patients followed either the HCR or LCR protocol. The HCR protocol was based on treatment with an initial caloric intake of 1800 kcal per day or 1600 kcal for patients at increased risk of RFS (<70% IBW, BMI < 15, or with electrolyte abnormalities) and children less than 12 years of age. The meal plans were increased by an average of 200–300 kcal per day to reach a final meal plan of 3000–3600 kcal per day. The LCR protocol provided patients with 1400–1500 calories per day initially and increased by 200–300 kcal every 3 days until a goal of 2600–2700 kcal was reached. Analysis of the results showed that patients in the HCR group presented a faster increase in BMI than patients in the LCR group. There was an increased ratio of mild hypophosphatemia in the HCR group, with no difference in the ratio of moderate and severe hypophosphatemia [16].

Interestingly, the hypothesis regarding the effect of rate of weight gain and caloric content of the diet on the likelihood of hypophosphatemia, which is described as a component of the RFS, has been challenged [17,18,19,20,21,22]. Yamazaki demonstrated that it is not the high-calorie diet but the higher % carbohydrate diet that increases the risk of refeeding hypophosphatemia (RH) in AN patients. In multivariate logistic regression analysis, the percentage of carbohydrate in total energy intake, higher than the cut-off point (58.4%), was significantly associated with RH incidence, even after adjusting for variables such as age and BMI, as well as average daily caloric intake. The group of patients who developed RH consumed more carbohydrates than the group without RH, with a lower average daily caloric intake (there was no difference in average daily caloric intake per body weight. To date, this is one of the few studies exploring the topic of the effect of dietary composition on weight restoration and the risk of complications during this process [23]. Also, the review by O’Connor et al. confirmed that the risk of RFS may depend not only on the caloric content of meals but also the protein/fat/carbohydrate ratio [24]. Recent studies show that high urea nitrogen to creatinine ratio and hypokalemia predispose to electrolyte disturbances and somatic complications [25,26]. Furthermore, a caloric supply below daily requirements during the initial hospitalization period predisposes to weight loss, potentially contributing to increased cardiovascular risk and prolonged hospitalization.

Haynos analyzed whether a structured, “rigid” weight restoration plan for hospitalized patients is more beneficial than an individualized nutritional rehabilitation plan. Both groups of patients with AN began hospitalization with an initial caloric menu of 1800kcal. The individualized approach group had calorie increases only if the weight did not gain at the target rate of 0.35 kg every 2 days between regular weight checks. The group with the standardized approach, starting from the 7th day of hospitalization, had their caloric intake regularly increased by 400 kcal every 2nd day until they reached the target maximum caloric intake of 3700 kcal/day. The experiment proved that the standardized rehabilitation plan implied curvilinear weight gain until reaching a maximum of about 2 kg/week in the third week of hospitalization, after which the rate of weight gain began to stabilize. On the other hand, patients receiving the individualized nutrition program gained weight in a fairly gradual manner, to a maximum weekly weight gain of 1.59 kg/week in the fourth week of hospitalization. Importantly, life-threatening complications such as RFS were not observed in either group, and the group with the standardized feeding plan was significantly less likely to require limiting physical activity. This is an important report also from an organizational and economic perspective—faster weight normalization is associated with lower hospital costs [10].

One approach to feeding hospitalized patients may be nasogastric tube (NGT) feeding, but the opinions of researchers on when to incorporate this type of feeding vary considerably. Most programs only introduce an NGT when the patient is unable to eat full meals [27]. Attitudes of medical professionals vary—some view it as invasive, others view NGT feeding as a method that minimizes physical and psychological discomfort during the initial stages of nutritional restitution [28]. The Agostino’s study proves that NGT feeding with an initial higher caloric intake leads to better early weight gain and shorter hospitalization, without an increased risk of RFS. The author compared two groups of patients—those fed standard meals and those who had NGT placed at the beginning of their stay. The NGT feeding protocol was based on an initial age-dependent daily caloric intake of 1500 kcal/day or 1800 kcal/day with Nutren Jr. fiber formula with 44% carbohydrate content. Calorie intake increased by 200 kcal/day until maximum calorie intake is reached. Once the patient’s nighttime heart rate normalized and the patient reached maximum caloric intake or after 7 days of NGT feeding, the patient was switched to traditional oral feeding. Patients fed standard meals initially consumed 1000–1200 kcal/day and had their caloric intake increased by 150 kcal/day until the estimated daily requirement was reached. The mean initial caloric intake in the standard-fed cohort was 1069 kcal/day and in the NGT group 1617 kcal/day. No differences in overall weight gain were observed between the two cohorts. In contrast, the length of hospitalization was significantly different between the two groups, averaging 50.9 days in the standard group and 33.8 days in the NGT group. In addition, there was no statistical difference in the rate of rehospitalization within six months after discharge, in the rate of gastrointestinal complications requiring medical intervention [29].

Undoubtedly, many new data may be provided in the future by an ongoing project in which researchers are to compare the efficacy and safety of a low-carbohydrate enteral mix (28% carbohydrate) and a standard enteral mix (54% carbohydrate) [30].

## 3. Fatty Acids

A 2019 study found that patients with AN have dysregulated fatty acid (FAs) profiles and lipid metabolism. Differences in patients with AN compared with healthy controls demonstrate metabolic aberrations: a markedly different fasting FAs pattern and significant decreases in fasting FA ratios compared to controls. Three of the four FAs with elevated levels in AN belong to the *n*-3 polyunsaturated family, suggesting that the metabolic deregulation in AN is due to a relative increase in *n*-3 polyunsaturated fatty acids (PUFAs). After the meal, FAs partially normalized and were no longer significantly different from the concentrations observed in healthy control subjects, except for alpha-linoleic acid (ALA), suggesting that a single meal can “activate” lipid metabolism in AN. Hence, we conclude that lipid dysregulation plays a significant role in modulating the onset and maintenance of AN symptoms. The clinical improvement may not only be due to an increase in circulating *n*-3 FAs, but also to synergistic effects with treatment or an overall increase in calories during treatment sessions. More research is needed to clarify the mechanism by which *n*-3 acid dysregulation affects AN and to explain how diet and standard treatment synergistically normalize this dysregulation to improve positive AN outcomes [31]. Seitz points out that dietary supplements with already defined effects on gut microflora, such as omega-3 FAs, are another promising research topic for treating AN [32]. Despite many studies demonstrating the beneficial effects of PUFAs on various diseases and inflammatory conditions, comprehensive studies on the role of PUFAs in AN risk and disease course are lacking. This group includes the “essential fatty acids” linoleic (*n*-6) and alpha-linolenic *n*-3 (*n*-3), which are precursors for eicosapentaenoic acid *n*-3 (EPA), docosahexaenoic acid (DHA) and arachidonic acid *n*-6 (ARA). While a small number of studies suggest that *n*-3 PUFAs may alleviate AN symptoms, the association between AN and specific PUFAs is inconsistent, and the mechanisms by which PUFAs may benefit AN are poorly understood. Patients with AN not only show a different pattern of PUFAs concentrations compared to healthy controls but also diverse postprandial *n*-6 ARA metabolism [33,34].

PUFAs supplementation has shown benefits in improving body weight in several serious conditions, including cancer [35,36]. Studies have shown that *n*-3 PUFAs supplementation improves AN by inhibiting cytokine production (as tumor necrosis factor (TNF-a) and interleukin IL-1b) and promoting the release of hypothalamic orexigenic peptides and neurotransmitters (as neuropeptide Y (NPY), a-melanocyte stimulating hormone (a-MSH) and serotonin) [33,37]. Although patients with AN do not experience appetite loss as do cancer patients, the striking ability to resist cravings and restrict food results in rapid weight loss accompanied by low-grade inflammation. The aversion to high-fat foods reported by AN may have a psychological basis in part, but often complaints of discomfort after eating suggest that certain nutrients in food may adversely affect the risk of developing AN. This hypothesis is further supported by the finding that AN patients reporting a particular fear of high-fat meals (fat-phobic-AN) reported more episodes and more severe gastrointestinal symptoms compared to patients who did not exhibit such severe fear (nonfat-phobic-AN) [38]. Eating behavior and food choice in AN are characterized by insufficient intake and a strong preference for low-fat foods [39].

Baskaran’s prospective study analyzed the food diaries of female patients with AN. Participants who gained weight at the 6–12 month follow-up (a 10% increase in BMI from baseline) took in a relatively lower percentage of all calories from protein and a higher percentage from fat compared to patients who did not gain 10% of their body weight. Importantly, there was no difference between the two groups in terms of total caloric intake. BMI during the follow-up period was positively associated with the percentage of total calories obtained from fat, MUFAs (monounsaturated fatty acids), and PUFAs; in particular, an increase in PUFAs intake during follow-up was a predictor of weight gain. No differences in absolute carbohydrate or fiber intake were observed between AN groups who gained or did not gain weight during follow-up. Crucially, body composition measurements were not significantly different between girls who did not achieve weight gain and those whose BMI increased by more than 10%. As expected at follow-up, the second group had significantly higher BMI, percent body fat, lean body mass, and fat mass than the first one [40].

Since correcting an imbalanced nutritional state is a necessary step to restore weight and maintain a healthy weight in AN, PUFAs imbalance or deficiency is an interesting direction for future AN research. At the same time, double-blind studies showed that PUFAs supplementation did not affect anxiety, depression, obsessive-compulsive symptom severity, or eating disorder scores of AN patients, compared to the AN group receiving placebo [41,42].

These findings suggest that the mechanism underlying the clinical benefits of *n*-3 PUFAs in other disorders may also be relevant to the treatment of AN. *n*-3 PUFAs have been found to be beneficial in the treatment of a number of mental illnesses and in promoting weight gain in a number of chronic diseases that lead to cachexia There is potential for *n*-3 PUFAs as adjunctive treatment to improve body condition and psychological symptoms in AN. Because of its clinical utility, a high-powered randomized clinical trial to test the efficacy of specific PUFA formulations in the treatment of AN is of primary importance, because then we will be in a better position to further develop personalized PUFA formulation as a novel treatment. Yehuda postulates that PUFAs supplementation may be beneficial in the treatment of AN for yet another reason—the stabilizing effect of these acids on neuronal membrane fluidity index and improvement of cognitive function, and lowering of homocysteine levels [43,44,45,46].

## 4. Vitamins and Microelements

The prevalence of micro-and macronutrient deficiencies and their clinical implications in patients with AN are poorly understood, yet some of the clinical symptoms reported by patients may be caused by reduced levels of vitamins and minerals. These include sensory neuropathy as a result of vitamin B12 deficiency, neuropsychological symptoms and emotional lability, and features of Wernicke syndrome from vitamin B1 deficiency [47,48]. In Hanachi’s study the most common deficiencies were reductions in zinc, vitamin D, copper, selenium, vitamins B1, folic acid (B9), and B12. At the same time, 28% of patients were deficient in a single component on laboratory tests, and 33% were deficient in two components. Only 7% of patients did not present macro-and micronutrient deficiencies [49]. As demonstrated by the authors of the largest analysis to date of the biochemical components of the blood of patients with AN, the micronutrient status must be monitored and supplemented to avoid deficiencies that can induce clinical symptoms in the therapeutic process. In trials, on average, half of the patients present vitamin D deficiency [49,50]. A large meta-analysis of the Veronesse study found that although patients with AN reported similar vitamin D intakes compared with healthy controls, they had significantly lower levels of 25OH-D and 1,25OH-D on laboratory tests. The cholecalciferol supplementation introduced in this group fully normalized serum vitamin D levels [51]. There is a strong association between vitamin D deficiency and bone mineral density, immune response and depressive symptoms [50,52]. Analyses showed that vitamin D supplementation resulted in an improved emotional state in patients with the major depressive disorder without producing a similar effect in healthy individuals, which the authors explain as a “floor effect”—the level of negative emotions in healthy people is so low that any improvement induced by supplementation may be difficult to assess statistically [53,54]. Insufficient levels of the vitamin also correlated with the risk of anxiety symptoms [55]. This is important in relation to the fact that the lifetime prevalence of generalized anxiety syndrome or depressive episodes in patients with eating disorders is very high [56,57]. Therefore, it seems that vitamin D supplementation should be provided to all undernourished patients with AN [50].

The significance of copper (Cu) deficiency and its consequences in patients with AN is unclear. Theoretically, Cu deficiency may contribute to increased bone demineralization and blood morphologic abnormalities [58,59]. Cu is essential for proper Fe absorption, red blood cell formation, and cardiovascular integrity [60]. The role of Cu in various metabolic diseases is widespread, both as a cofactor of mitochondrial proteins and as a regulator of neurotransmission processes, antioxidant protection, and a determinant of dopamine β-oxidase, superoxide dismutase (SOD), and tyrosinase activity [61]. A transcranial direct current brain stimulation (tDCS) study conducted on rodents found changes in Cu levels in brain tissue, which the authors postulate as one of the factors regulating the hunger and satiety centers that modulate eating behavior [62].

The suggested effect of zinc (Zn) on psychopathological symptoms has been described in numerous studies on depressive disorders, schizophrenia, irritability, and ADHD [63]. In addition, Zn has been shown to affect glutaminergic and serotonergic regulation as well as neuroproliferation and systemic anti-inflammatory effects [64,65,66,67]. In a study of patients with eating disorders, Zn levels were found to be reduced [68]. Rosenblum also described a case of Zn deficiency-induced cardiomyopathy in a patient diagnosed with acute phase AN in whom other possible etiopathogenetic factors were excluded. The patient presented with reduced Zn levels on laboratory tests and their heart failure improved after Zn supplementation. The authors concluded that when inflammation occurs, plasma Zn deficiency may lead to a relative deficiency of antioxidant enzymes, ultimately contributing to apoptosis and myocardial necrosis [69]. Zn plays a key role in regulating excitotoxicity through negative feedback (e.g., down-regulation or reduced sensitivity of NMDA receptors) [70]. Decreased Zn levels have also been associated with neuropsychiatric symptoms such as cognitive impairment, mood disorders, psychosis, and symptoms of major depression [66,71]. Numerous studies have shown that a deficiency of this element can promote the development of AN, anxiety, and depressive symptoms [72,73]. In the Zepf study, Zn concentrations were higher in patients in remission of AN compared to patients in the acute phase of the disease and also the control group [74].

The 12-week Zn supplementation also increased serum BDNF levels and reduced depressive symptoms. With regard to the association of reduced levels of this factor in individuals with eating disorders proven in multiple publications and reports that BDNF is positively associated with body weight in women, this element is gaining increasing scientific interest [75,76]. Studies indicate that reduced Zn concentrations correlate with weight loss and a study published several years ago on a Zn-supplemented AN group showed a two-fold increase in BMI [77,78,79].

Furthermore, Hermens postulates the potential for Zn to be used in ketamine therapy, as a treatment option for AN, with respect to restoring glutamatergic neurotransmission, generating key symptoms of AN, including changes in interoceptive awareness, metastability of attention, or compulsive behavior [80,81,82,83].

## 5. Microbiota and Its Modulations

The gut microbiota is an increasingly widely recognized factor in the etiopathogenesis and maintenance of AN symptoms. The gut microbiome plays an important role in regulating mood, appetite, and metabolic processes [84,85,86]. In addition, it interacts with biochemical, neuroimmune, and neurotransmitter pathways of AN patients through the gut-brain axis [85,87]. Ongoing studies of the microbiota of people with AN over the past several years are uncovering a growing picture of changes in gut bacterial composition: A reduction in total bacterial numbers, including Bacteroides fragilis, Lactobacillus plantarum and Roseburia, and carbohydrate-fermenting Ruminococcus along with a subsequent reduction in propionic and butyric acids [88,89,90,91]. Importantly, butyrate levels have been described to negatively correlate with anxiety and depression, which may explain the increased levels of anxiety in individuals with AN, as well as impaired gut physiology and motility [89,90,92].

Given the available data and the lack of good biomarkers for AN, microbiota may be a good point of intervention in the management of patients with AN. There is a growing interest in the immunomodulatory role of prebiotics and probiotics in the treatment of mood disorders, which, with respect to the frequency of these symptoms in AN, may represent an important aspect of therapeutic interactions. Butyrate-producing Roseburia bacteria have been suggested as candidates for probiotic intervention studies in AN [89,93]. A meta-analysis of clinical trials determining the role of prebiotics and probiotics in modulating depressive and anxiety symptoms showed that prebiotic supplementation has no effect on psychiatric disorders, whereas probiotics have antidepressant and anti-anxiety effects [94]. At the same time, as the authors themselves point out, despite the great interest in the microbiome in relation to depression and anxiety, only a few studies have looked at patients with clinical symptoms of depressive disorders, and the study group sizes were often not large enough.

Tetyana Rocks postulates the use of fermented foods in modulating gut microflora in the nutritional rehabilitation of AN patients. It has been shown to alter the composition and function of the human gut microbiota and is therefore presumed to have a neuroprotective role [95,96]. Fermented foods can, by modulating the gut microbiota, potentially reduce oxidative stress, which is all the more important because it causes cognitive impairment by inducing neuronal cell death [97]. Fermented foods can also modulate the production of neurotransmitters e.g., BDNF, glutamate, GABA and serotonin, as well as insulin-like growth factor I [98]. For AN patients, fermented foods are also an energy- and nutrient-rich food option to support nutritional rehabilitation. Therefore, their use in nutritional treatment protocols represents a real opportunity to modulate metabolic and immunological processes, but also to reduce psychopathological symptoms and cognitive impairment.

To date, there is no evidence to support the recommendation of supplementation with prebiotics or probiotics. However, rapidly emerging evidence suggests that non-digestible carbohydrates and prebiotic foods play an important role in producing short-chain fatty acids, as well as increasing levels of beneficial intestinal Bifidobacteria and lactic acid bacteria [19,99,100] Furthermore, based on animal studies, it is known that Bifidobacteria supplementation leads to increased levels of the serotonergic precursor tryptophan, suggesting a possible role for probiotics in modulating neurotransmitter levels [101,102]. Dietary sources of prebiotics include indigestible carbohydrates in the form of rye, wheat, barley, oats, and legumes, and non-digestible oligosaccharides such as inulin, fructans, polydextrose, fructo-oligosaccharides, and galacto-oligosaccharides [103].

Nutritional rehabilitation of AN requires a high-calorie diet, but as postulated by Anu Ruusunen, a combination of high fat and high fiber may be beneficial. This composition may promote the composition of the gut microbiome similar to that of a healthy population. According to these recommendations, it is important that the fat component of the diet comes from unsaturated fats rather than saturated fats [104]. At the same time, it is worth noting that exposure to a wide variety of foods is an important part of treating AN. This may also include foods that are considered unhealthy, such as those rich in saturated fat or simple sugars but are psychologically beneficial as a treatment factor for food anxiety. Therefore, some of these foods should remain part of nutritional rehabilitation. Promoting a fiber-rich diet during nutritional rehabilitation in AN can be difficult in practice because additional fiber intake means increased amounts of low-energy foods and increased feelings of satiety. Therefore, it is especially important to strive to include small amounts of fiber in every meal or snack provided. This stepwise approach will also be important in treating bowel symptoms in patients with AN. With increasing evidence in gut microbiota research, particularly in AN, personalized re-feeding protocols may ultimately be necessary, given the demonstrated inter-personal differences in the human gut microbiome.

Weight restoration in AN in hospitals is accomplished with foods administered as part of a standard or individualized meal plan, as well as with medical foods (Nutridrink) and/or nutrition through NGT [105]. Exclusive enteral feeding has been shown to alter the gut microbiota and lead to decreased production of short-chain fatty acids which is at least partially explained by the low fiber content of enteral formulas [106]. Parenteral nutrition (PN) is not recommended because of the risk of complications, including disseminated intravascular coagulation (DIC), edema, sepsis, and elevated transaminases. It has been postulated that PN disrupts the normal microbiome and may explain the observed impairment of immune and epithelial barrier function observed during long-term intravenous feeding The only indication for PN in AN patients to date is the lack of other alternatives to restore weight in patients whose complications preclude enteral nutrition [9]. Meanwhile, published studies describing the gut microflora of patients with AN usually did not use NGT or PN in the study populations. Furthermore, also studies describing the effects of total enteral nutrition or PN did not include AN patients [107,108]. Therefore, the potentially important effects of these nutritional rehabilitation methods on gut microflora remain unknown.

It is encouraged that future nutritional rehabilitation protocols in AN include higher and more diverse fiber content and prebiotic foods. This may benefit the gut microbiota and its metabolites and prevent gastrointestinal sequelae [89,93]. Similarly, future research is needed to explore the potential role of probiotic foods, fermented foods, or supplements as part of nutritional rehabilitation procedures. These insights may provide new clues for modulating gut microflora to improve treatment outcomes.

## 6. Conclusions

Restoration of proper nutritional status is the primary goal in AN treatment. In the process of nutritional rehabilitation, there are several methods described, based, for example, on oral nutrition with standard meals, medical nutrition in liquid form, nutrition other than oral (NGT and enteral probes, PN), and the use of supplementation. Based on the results of the above literature review, it is apparent that there is limited evidence for the superiority of a single effective feeding method in the nutritional rehabilitation of patients with AN. The current publications provide many clues for further research as the priority in the field of eating disorder research and AN is to advance knowledge and understanding of nutritional rehabilitation in AN.

Although many studies are available on this topic, no consistency has been observed in comparing different nutritional protocols. Most of the studies conducted have focused solely on the perspective of adult patients, whereas both the microbiota and phenotype of AN may differ in these age groups. Additionally, the experiments regarding the caloric content of the menus were conducted with adults, which does not allow for direct translation of the recommendations to the adolescent population. Further research is needed to determine the impact of standardized caloric recommendations in different clinical populations. Another limitation of the current literature is the relatively small number of randomized and double-blind studies. In addition, by observing the durations of the cited study programs, we suggest that future studies should be of similar duration and include follow-up of patients after treatment and assessment of the durability of weight normalization.

## Data Availability

Not applicable.

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
