# Peer review of "Anorexia Nervosa—What Has Changed in the State of Knowledge about Nutritional Rehabilitation for Patients over the Past 10 Years? A Review of Literature"

_nutrients, 2021, doi:10.3390/nu13113819_

Round 1
Reviewer 1 Report
Comments for the Authors:
This paper by Jowik and collegues is a review of changes in nutritional practices for patients with Anorexia Nervosa over the past 10 years.
Specific Comments:
Title: OK as is
Abstract: OK as written
Introduction: OK as written
Calories and nutrients in the re-nutrition plan
P 2 Line 96: There is something missing after the period and before demonstrated.
P3 Line 105: I suggest a modification: “Also, the review by O’Connor et al (2011) confirmed…..”
P 3 Line 134: Why the blank space at the end of the line? It is not a new paragraph.
P 4 Line 149: Why the big gap between the end of one and beginning of next sentence?
Fatty Acids
P 4 Line 156: Deregulated doesn’t fit here. Please select another word or phrase.
P 4 Line 158: Please describe the “aberrations” in more detail.
P 4 Line 158: I don’t follow your reasoning that fatty acids modulate the onset and continuation of AN. Please expand.
P 4 Line 175: Which peptides and neurotransmitters? Please be specific.
P 4 Line 189: Were there any differences in change in body composition with these different diets?
Vitamins and microelements
P 5 Line 217: By elements, do you mean minerals? Please reword
P 5 Line 221: Please replace B9 with folate by which is more commonly known.
P 5 Line 237: The phrase “reduction effect” is peculiar in English. Please reword.
P 5 Line 240: Please make episode plural – episodes
MIcrobioata and its Modulations
Line 6 P 292: please add acid after propionic and butyric or make them into propionate and butyrate.
P 7 Line 352: Why the gap at the end of the sentence?
P 7 Line 354: Is this word spelled correctly, and what does it mean? Transaminasemia
Conclusions: OK as written
Author Response
Dear Sir,
I sincerely thank you for your evaluation of the manuscript and all the constructive comments to which I have tried to apply:
-P 2 Line 96: Yamazaki demonstrated that
-P3 Line 105: Modified as suggested: Also, the review by O’Connor et al confirmed…..
-P 3 Line 134: corrected
-P 4 Line 149: corrected
-P 4 Line 156: changed to dysregulated
-P 4 Line 158: issue developed: a markedly different fasting FAs pattern and significant decreases in fasting FA ratios compared to controls. Three of the four FAs with elevated levels in AN belong to the n- 3 polyunsaturated family, suggesting that the metabolic deregulation in AN is due to a relative increase in n-3 polyunsaturated fatty acids (PUFAs). After the meal, FAs partially normalized and were no longer significantly different from the concentrations observed in healthy control subjects, except for alpha-linoleic acid (ALA), suggesting that a single meal can "activate" lipid metabolism in AN. Hence, we conclude that lipid dysregulation plays a significant role in modulating the onset and maintenance of AN symptoms. The clinical improvement may not only be due to an increase in circulating n-3 FAs, but also to synergistic effects with treatment or an overall increase in calories during treatment sessions. More research is needed to clarify the mechanism by which n-3 acid dysregulation affects AN and to explain how diet and standard treatment synergistically normalize this dysregulation to improve positive AN outcomes
-P 4 Line 175: developed: cytokine production (as tumor necrosis factor (TNF- a) and interleukin IL-1b) and promoting the release of hypothalamic orexigenic peptides and neurotransmitters (as neuropeptide Y (NPY), a-melanocyte stimulating hormone (a-MSH) and serotonin)
-P 4 Line 189: developed: Crucially, body composition measurements were not significantly different between girls who did not achieve weight gain and those whose BMI increased by more than 10%. As expected at follow-up, the second group had significantly higher BMI, percent body fat, lean body mass, and fat mass than the first one
-P 5 Line 217: minerals
-P5 Line 221: folic acid
-P5 Line 237: the level of negative emotions in healthy people is so low that any improvement induced by supplementation may be difficult to assess statistically
-P5 Line 240: converted into plural
-P6 Line 292: added propionic and butyric acids
-P7 Line 352: removed gap
-P7 Line 354: modified to transaminasemia
Reviewer 2 Report
This review of literature collected and summarized the recent evidence on the nutritional rehabilitation approach and its effectiveness in anorexia nervosa.
I found this work very interesting and useful for further studies because it allows having a summary of the state of the art on the topic. In addition, deeping nutritional and rehabilitation aspects in eating disorders is important, because as expressed by the authors they represent a significant public health problem and a cost to the health care systems.
As a suggestion: to better describe the limitations highlighted by the different studies regarding the different nutritional intervention strategies.
Author Response
Dear Sir or Madam,
Thank you very much for reviewing our publication. In the conclusions of the paper, I tried to highlight the main limitations of the experiments so far and the directions for future research on the nutritional rehabilitation of patients with AN.
Conclusions:
Restoration of proper nutritional status is the primary goal in AN treatment. In the process of nutritional rehabilitation, there are several methods described, based, for example, on oral nutrition with standard meals, medical nutrition in liquid form, nutrition other than oral (NGT and enteral probes, PN), and the use of supplementation. Based on the results of the above literature review, it is apparent that there is limited evidence for the superiority of a single effective feeding method in the nutritional rehabilitation of patients with AN. The current publications provide many clues for further research as the priority in the field of eating disorder research and AN is to advance knowledge and understanding of nutritional rehabilitation in AN. Although many studies are available on this topic, no consistency has been observed in comparing different nutritional protocols. Most of the studies conducted have focused solely on the perspective of adult patients, whereas both the microbiota and phenotype of AN may differ in these age groups. Additionally, the experiments regarding the caloric content of the menus were conducted with adults, which does not allow for direct translation of the recommendations to the adolescent population. Further research is needed to determine the impact of standardized caloric recommendations in different clinical populations. Another limitation of the current literature is the relatively small number of randomized and double-blind studies. In addition, by observing the durations of the cited study programs, we suggest that future studies should be of similar duration and include follow-up of patients after treatment and assessment of the durability of weight normalization.